# Upgrading the Bioactive Potential of Hazelnut Oil Cake by *Aspergillus oryzae* under Solid-State Fermentation

**DOI:** 10.3390/molecules29174237

**Published:** 2024-09-06

**Authors:** Melike Beyza Ozdemir, Elif Kılıçarslan, Hande Demir, Esra Koca, Pelin Salum, Serap Berktaş, Mustafa Çam, Zafer Erbay, Levent Yurdaer Aydemir

**Affiliations:** 1Department of Food Engineering, Adana Alparslan Türkeş Science and Technology University, Adana 01250, Türkiye; dyt.melikebeyzaozdemir@gmail.com (M.B.O.); esrakoca.tr@outlook.com (E.K.); pelinsalum@ymail.com (P.S.); zerbay@atu.edu.tr (Z.E.); 2Graduate School of Natural and Applied Sciences, Osmaniye Korkut Ata University, Osmaniye 80000, Türkiye; elifkilicarslan93@gmail.com; 3Department of Food Engineering, Osmaniye Korkut Ata University, Osmaniye 80000, Türkiye; 4Department of Food Engineering, Erciyes University, Kayseri 38280, Türkiye; berktaserap@gmail.com (S.B.); mcam@erciyes.edu.tr (M.Ç.)

**Keywords:** hazelnut oil cake, solid state fermentation, *Aspergillus oryzae*, bioactive properties, functional properties

## Abstract

Hazelnut oil cake (HOC) has the potential to be bioactive component source. Therefore, HOC was processed with a solid-state fermentation (SSF) by *Aspergillus oryzae* with two steps optimization: Plackett–Burman and Box–Behnken design. The variables were the initial moisture content (X_1_: 30–50%), incubation temperature (X_2_: 26–37 °C), and time (X_3_: 3–5 days), and the response was total peptide content (TPC). The fermented HOC (FHOC) was darker with higher protein, oil, and ash but lower carbohydrate content than HOC. The FHOC had 6.1% more essential amino acid and benzaldehyde comprised 48.8% of determined volatile compounds. Fermentation provided 14 times higher TPC (462.37 mg tryptone/g) and higher phenolic content as 3.5, 48, and 7 times in aqueous, methanolic, and 80% aqueous methanolic extract in FHOC, respectively. FHOC showed higher antioxidant as ABTS^+^ (75.61 µmol Trolox/g), DPPH (14.09 µmol Trolox/g), and OH (265 mg ascorbic acid/g) radical scavenging, and α-glucosidase inhibition, whereas HOC had more angiotensin converting enzyme inhibition. HOC showed better water absorption while FHOC had better oil absorption activity. Both cakes had similar foaming and emulsifying activity; however, FHOC produced more stable foams and emulsions. SSF at lab-scale yielded more bioactive component with better functionality in FHOC.

## 1. Introduction

Oilseed cakes such as soy bean, maize, sunflower, and cottonseed are widely used in animal nutrition due to their high phenolic and protein, and balanced amino acid, content. However, the potential of their protein content for human nutrition has not been sufficiently unlocked yet [1,2,3]. In addition to its protein content, oilseed cakes can be regarded as a bioactive component source, due to their oleochemicals, phytochemicals, fiber, minerals, and many other health-related ingredients with strong antioxidant properties [4]. Among industrially produced oilseed cakes, hazelnut oil cake (HOC), which is released as a by-product during the production of hazelnut oil, is also preferred in animal nutrition due to its low cost, high protein (40–55%) and carbohydrate (35–50%) content, and good protein digestibility (80–90%). In addition to its use in animal feed, the studies show that hazelnut oil cake can be a good candidate for human consumption due to its nutritional content [2,5,6]. The potential of HOC has been demonstrated by studies in the literature. According to the Phenol-Explorer database, hazelnut ranks 21st among 452 food products (fruits, vegetables, seeds, nuts, cereals, oils, and beverages) with 495 mg phenolic content (determined by chromatographic method) per 100 g among polyphenol-rich foods [7]. HOC contains a significant number of phenolic compounds, although less than the shell and membrane; these values vary between 0.53–15.96 mg GAE (gallic acid equivalent)/g extract [8,9,10]. Among the phenolic compounds, the most commonly detected compounds in HOC are quinic acid, quercetin-3-O-rhamnoside, (+)-catechin, catechol, glansreginin A, glansreginin B, syringic acid hexose esters, procyanidin dimer, trimer, and tetramers [7].

The hazelnut oil cake obtained after cold-pressed oil production was processed to produce hazelnut protein, hazelnut beverage, kefir, edible film, chocolate, chocolate spread, additive for wheat dough, yoghurt-like product, meatball, frankfurter type beef sausages, and extruded snack products in food science studies [2,4,6,8,11,12,13].

According to data from FAOSTAT 2022, hazelnut production in the world was estimated as 1.2 million tons, and approximately 65% of the world hazelnut production took place in Türkiye, which was followed by Italy, Azerbaijan, and the USA Most of the produced hazelnut is consumed as roasted snack and hazelnut paste; the rest of the limited amount of the nuts (10%) is used to produce hazelnut oil, predominantly cold-pressed. Therefore, Türkiye can be regarded as a sustainable source for hazelnut oil cake which deserves valorization for implementation in the food, cosmetic, and pharmaceutical industries. These industries are required to have proteins, bioactive peptides, essential amino acids, phenolic compounds, dietary fibers, prebiotics, vitamins, and minerals to increase nutritional content, to support well-being and health, to provide food safety, and to sustain product quality. Those bioactive components can be extracted from oil seed cakes by various techniques. The high content protein extracts are mostly obtained by acid/alkali extraction and isoelectric point/salt precipitation after mechanical treatments; however, those methods generally give low extraction yields due to complex formation between proteins and carbohydrates, phenolic compounds, and other non-protein components in the medium. This complexation makes it difficult for proteins to dissolve in water and subsequently to precipitate and be purified [2,6,14]. Those proteins have valuable nutrition content; however, they have limited bioactive properties which can be improved by chemical or enzymatic hydrolysis that produce protein hydrolysates (as bioactive peptides) [14]. For phenolic content extraction, various solvents such as water, ethanol, methanol, isopropanol, ethyl acetate, hexane, deep eutectic solvents, and other organic solvents are used. Among the solvents, water generally gives the lowest extraction yield, and deep eutectic solvents are difficult to prepare, with relatively high cost. Although organic solvents provide higher extraction yields, they may have toxicity, cause harmful effects to environment, and need extra recovery and disposal steps [15,16]. For a greener approach, ultrasound assistance, microwave treatment, enzymatic hydrolyzation, high pressure homogenization, and supercritical fluid extraction techniques may also be used to increase phenolic extraction yield; however, they may have some drawbacks regarding cost, industrial scale adaptability, being sophisticated, and operational difficulties [13,15,16,17,18]. Similar to protein extraction, the lignocellulosic structure in the cake prevents higher phenolic compound extraction yield due to complexation with carbohydrates and proteins.

The fermentation can be regarded as a suitable solution for bioactive component extraction from oil seed cakes because microorganism may have the ability to hydrolyze the complex carbohydrates and proteins in plant tissues, resulting in free phenolic compounds, proteins, peptides, and amino acids. The microbial activity may also contribute the bioactive component and protein formation in the medium [19,20]. Although different types of fermentation techniques have been applied to the oilseeds, the solid-state fermentation (SSF) which takes place on a solid matrix with little or no free water present is an alternative way due to being more practical and environmentally friendly technology. The formation of bioactive components by SSF depends on process conditions such as initial moisture content, incubation temperature, and incubation time, which are critical parameters [21]. Research in the SSF area is generally focused on commercial enzyme production. In addition to this, there have also been many studies seeking to increase the amount of free phenolic and protein content, to produce bioactive peptides, and to eliminate antinutritional factors from fermented products [22,23,24,25,26,27,28]. When black bean flour, soybean cake, peanut cake, flaxseed cake, and hazelnut cake were fermented using various microorganisms for these purposes, significant increases in the protein content of fermented products were determined [23,29,30,31] (*p* < 0.05). In addition, in the fermentation studies of various legumes, oilseed meals, wheat, and hemp by *Bacillus* species, *Aspergillus* species, *Lactobacillus* species, *Rhizopus oligoporus*, Bifidobacterium, and *Pleurotus ostreatus*, significant changes occurred in soluble phenolic compounds, protein content, and antioxidant activities of the fermented products, while significant improvements were observed in the antihypertensive, antidiabetic, and anticancer activities of some of the products [20,22,23,24,29,32,33,34,35,36,37]. Among the SSF studies in the literature, there are limited research where fermentation of HOC was studied. In these studies, ethanol and various enzymes were produced by fermenting the outer hard and inner soft shells of hazelnut with SSF [38,39,40,41,42,43,44,45,46]. Similarly, studies on the production of alpha amylase using *Bacillus amyloliquefaciens* and increasing the protein content of the meal for animal nutrition using *Aspergillus niger* ATCC 9142 were also carried out [47,48]. In these studies, it was observed that while the hemicellulose content of HOC decreased with fermentation, the protein content increased by 31% (*p* < 0.001). These studies show that HOC is a suitable material for SSF applications and the type of the microorganism in SSF is one of the important factors on the characteristics of the fermented product. When the fermentation studies on HOC were reviewed, no study was found to increase the bioactive component content of HOC using *Aspergillus oryzae*. *A. oryzae* is a mold that has produced fermented products such as koji, sake, shoyu, and miso in the Far East since ancient times [49,50]. It is in GRAS (generally accepted as safe) practices and is in the MFC (microbial food cultures) class. Therefore, there is no deficiency in its consumption with the fermented product after fermentation. This also eliminates the decomposition of microorganisms in the downstream processes [51]. As a non-pathogenic microorganism, it does not produce aflatoxin [49]. Many hydrolytic enzymes, such as α-amylase, β-glucosidase, protease, and lipase, are secreted by *A. oryzae* during fermentation [52,53,54]. Studies have shown that peptides with various bioactive properties can be produced from different protein sources after hydrolysis by the proteases of *A. oryzae* [26,27,28,55,56,57]. These studies clearly demonstrate that *A. oryzae* hydrolyzes proteins with high proteolytic activity analysis, which was one of the motivations for the current study.

Functional properties provide insight into the usability of cereal-based flours, protein extracts, and similar carbohydrate-protein-based powder blends in industrial applications. For this reason, many studies on plant proteins or oilseed cakes have investigated properties such as oil and water absorption capacity, emulsifying activity, emulsion stability, foaming activity, and foam stability [2,58,59,60,61]. Water absorption/retention influences the texture, juiciness, and taste of food formulations, especially impacting the shelf-life of bakery products [59]. A high water absorption capacity (WAC) allows bakers to incorporate more water into doughs, enhancing handling characteristics and maintaining bread freshness. The improved ability of the meal to absorb and retain water and oil can enhance structural binding, flavor retention, mouthfeel, and reduce moisture and fat losses in food products [62]. The oil absorption capacity (OAC) relies on the physical entrapment of the oil molecules by a capillary-attraction process of the hydrophobic molecules such as proteins [2,62].

Therefore, in the current study, we aimed to produce fermented hazelnut oil cake with increased bioactivity so that the optimization of the SSF process conditions by *Aspergillus oryzae* using Box–Behnken design and response surface methodology was done and the FHOC was characterized to reveal its potential to be used as bioactive and functional food ingredient.

## 2. Results and Discussions

### 2.1. Optimization of SSF Conditions

The formation of bioactive components by fermentation depends on process conditions and determining the right conditions is very important for fermentation [21]. The literature survey showed that the initial moisture content (MC) and water activity of the environment, the selected microorganism, incubation time, and incubation temperature are the factors that have significant effects on the final product [63,64,65,66]. For this reason, the effects of initial moisture content, incubation temperature, and incubation time on total peptide content (TPC) of the fermented hazelnut oil cake (FHOC) after the SSF process were investigated using a Box–Behnken experimental design with 17 runs. The experimental results from all factor combinations and responses of each run are given in Table 1. 

The second-order polynomial regression equation showing the empirical relationship between the TPC and the independent variables in terms of coded levels of SSF conditions is given in Equation (1).
Total peptide content (mg tryptone eqv./g DM of solid substrate) = 279.0 + 66.9 · X_1_ − 24.2 · X_2_ + 77.5 · X_3_ − 14.1 · X_1_X_2_ + 52.9 · X_1_X_3_ − 47.6 · X_2_X_3_ − 33.3 · X_1_^2^ − 34.7 · X_2_^2^ − 67.6 · X_3_^2^(1)

Analysis of variances (ANOVA) based on the TPC was carried out to evaluate the significance and fitness of the model coefficients is summarized in Table 2. A large value of F and a small value of *p* (i.e., *p* < 0.05) indicate that the model is statistically significant when examining the results of ANOVA [67]. Results given in Table 2 show that the constructed model is significant (*p* < 0.05). In addition, main terms of initial moisture content and incubation time, their interaction, and quadratic term of incubation time were found to have significant (*p* < 0.05) effect on TPC response, whereas the incubation temperature did not have a significant (*p* < 0.05) impact on the TPC. A relatively high R^2^ (0.916) value indicated a good correlation between the experimental and predicted responses. Moreover, the predicted R^2^ (0.665) was found to be in reasonable agreement with the adjusted R^2^ (0.809) since the difference is less than 0.2.

The preparation of 3D plots as a function of two parameters while maintaining all other parameters at fixed levels is useful for comprehending the interaction effects of the two parameters [67]. The effects of interaction terms (X_1_X_2_, X_1_X_3_, X_2_X_3_) on TPC were visualized on response surface plots. Figure 1a–c clearly shows that at the lower levels of incubation temperature and higher levels of incubation time, TPC increased as the initial moisture content % increased from 30 to 70%. Fungal proteases are known to catalyze the hydrolysis reactions to obtain bioactive peptides [68]. There are numerous reports on the production of protease using fungal cultures under the SSF conditions. Most of them observed the optimum temperature range of 28 to 30 °C, similar to our findings [51]. Contrarily, de Castro and Sato (2014) observed an increase in the protease activity secreted by *A. oryzae* on wheat bran with increasing incubation temperature up to 55 °C [53]. However, they observed low enzymatic stability above 45 °C.

Initial moisture content below the optimum level may lead to low nutrient diffusion, microbial growth, enzyme stability, substrate swelling, and sporulation, whereas above this level the problems may be particle agglomeration, gas transfer limitation, and competition with bacteria. The moisture content required for the fungal growth or secretion of metabolites generally varies between 40–80%, yet it heavily depends on the nature of the substrate. The optimum moisture content levels reported for various strains and substrate pairs in SSF processes of protease production were 50% for wheat bran (*A. oryzae*) [69], 40% for Jatropha seed cake (*A. versicolor*) [70], 50% for wheat bran-Aspergillus isolate [71], and 70.5 for sugar cane bagasse (*A. niger*) [72]. In our study, it was observed that in order to have a high total peptide content, approximately 70% initial moisture content was required by using the hazelnut oil cake–*A. oryzae* combination with the SSF method (Figure 1a,b).

Composition of SSF media may change over time during fermentation. For this reason, the period for maximum production of target component should be determined and fermentation should be terminated quickly at that moment [73]. Otherwise, there may be risks such as contamination, increase in cost, and accumulation of inhibitory metabolites. De Castro and Sato (2014) observed that protease production by *A. oryzae* LBA 01 under SSF of cotton seed meal reached to its maximum level at the 72nd hour of the process [53]. Ibarruri et al. (2019) reported a maximum degree of hydrolysis level at the 96th hour of SSF by *Rhizopus* sp. incubated on brewer’s spent grain and they related this result to the maximum peptide yield [64]. As can be seen in Figure 1b,c, at 30% of initial moisture content and 40 °C of incubation temperature, TPC continuously increased until 3.8 days, then it decreased. According to Lu et al. (2022), the production of protease in the early stage of microbial growth may increase the peptide content [74]. However, at the later stage of SSF, the peptides will be further hydrolyzed into amino acids that may cause a decrease in the peptide content. This result also indicates the importance of optimizing the fermentation time (incubation time) as a significant factor.

### 2.2. Validation of Optimum Conditions

The fitness of the model equation (Equation (1)) for predicting optimum response value was investigated under the conditions given in Table 3. In all five runs, desirability values were 1.000. Experiments were carried out twice and average values were given. In the comparison of predicted vs. experimental values, calculated error percentages must be as low as possible [60]. Optimum conditions that maximize TPC were determined as initial moisture content of 69.8%, incubation temperature of 24.6 °C, and incubation time as 4.6 days with the least error percentage. Table 4 also shows that the limit of errors for the response is within a tolerable range, supporting the correctness of the established model (Equation (2)).

The optimum conditions obtained in this study resulted in the maximum TPC as 404.4 mg tryptone eqv./g dry matter (DM) of solid substrate. Unfermented (raw) HOC was determined to have 40.4 mg tryptone eqv./g DM of solid substrate, which means that SSF by *A. oryzae* under the optimized conditions achieved to increase TPC of HOC by 10-times.

Jiang et al. (2020) reported a comparable maximum bioactive peptide content of solid-state fermented (with *Bacillus subtilis* MTCC5480) corn gluten meal as 369.4 mg/gdp [75]. There is a relatively lower total peptide content recently reported after the SSF of some by-products. The maximum peptide content during the fermentation of soybean meal by *B. subtilis* was 135.6 mg/g [74]. The optimum conditions of SSF of okara/chicken feather powder by *B. licheniformis* revealed a total peptide content of 185.99 mg/g [76]. SSF of another *B. licheniformis* strain on brewer’s spent grain and soybean meal increased the peptide content of these co-substrates from 25.71 mg/g to 112.72 mg/g under optimal SSF conditions [77].

### 2.3. Characterization of HOC and FHOC

The color values of HOC was expressed in terms of L*, a* and b* as 81.4 ± 1.9, 1.7 ± 0.3 and 16.7 ± 0.6, respectively. The moisture content of HOC was 8.25 ± 0.35 % and water activity was 0.542 ± 0.001.

Statistical analysis (*t*-test, *p* ≤ 0.05) showed that there were significant differences in all general properties of HOC and FHOC (Table 4). The fermentation process caused an increase in protein, lipid, and ash contents of the FHOC, while it caused a decrease in dry matter and carbohydrate contents. The significant changes during fermentation might be related to microbial activity because microorganisms produce the energy they need during their growth and proliferation primarily by consuming carbohydrates in the HOC and they use the monomers in the medium to synthesize macromolecules such as protein and lipid. The increase in the amount of protein can be associated with the production of enzymes and cellular components in the microorganisms [78]. The similar increase in the protein content of fermented rapeseed meal by SSF using *Aspergillus oryzae* was observed in Dossou et al. (2019) [79]. It was also reported that the protein and ash content of fermented linseed meal increased, whereas the carbohydrate content decreased with fermentation process [80]. In addition to these, Feng et al. (2007) and Teng et al. (2012) observed an increase in both protein and fat content of the meal as a result of soybean meal fermentation [81,82].

At the end of the fermentation process, darkening in the color of the FHOC was observed. The main reasons for this color formation might be the presence of spores and mycelial structures formed during the development of molds and the presence of color pigments as the secondary metabolites required for microbial development. In addition, the tyrosinase activity shown by *Aspergillus oryzae* can catalyze enzymatic browning reactions and phenolic oxidation, leading to the formation of melanin and other color pigments [83]. These color changes are common in mold-used SSF systems, and studies suggesting microbial growth monitoring based on color change are also available in the literature [84].

### 2.4. Amino Acid Composition of HOC and FHOC

The amino acid content in HOC increased 24% after fermentation (Table 5). However, the increase in essential amino acid content was very limited (6.1%). Glutamic acid was the highest amount of amino acid found in HOC and FHOC, followed by aspartic acid and arginine, while valine and leucine content of the cakes ranked first among essential amino acids. The amino acid composition of HOC is consistent with other hazelnut oil cakes reported in the literature [85,86]. The fermentation process increased in threonine 21.91%, phenylalanine 18.63%, and leucine 10.17%, but decreased in lysine 25.13%. In addition, the only changes that were found to be statistically significant different were in the amounts of aspartic acid, glutamic acid, arginine, and phenylalanine. (*p* ≤ 0.05).

### 2.5. Volatile Compound Composition

The volatile compounds detected in HOC and FHOC are presented in Table 6. Thirteen compounds were detected in HOC and nineteen compounds in FHOC. The total determined volatile compound content in HOC was 6992.9 ± 3.0 μg/kg DM, while it was 2953.4 ± 11,359.1 ± 111.2 μg/kg DM in FHOC.

There were significant differences in the volatile composition of the headspace of the samples after fermentation. The only common compound detected among the samples was 2-ethyl-1-hexanol. This compound was also identified as a lipid oxidation product in hazelnut samples by Cialiè Rosso et al. (2018) [87]. The most predominant compound identified in the headspace of HOC was acetic acid (37.2%), an organic acid, followed by limonene (27.8%), a member of the terpene class, and hexanal (8.7%), a straight-chain aldehyde. The literature on the volatile compounds of HOC is limited. Burdack-Freitag and Schieberle (2012) reported that the highest amount of aroma active compound in raw hazelnuts was hexanal, followed by acetic acid [88]. In the same study, hexanal and acetic acid were also detected in roasted hazelnut paste samples. In another study, the same authors also detected these compounds in hazelnut and hazelnut paste [88]. Çam (2009) examined the presence of hexanal and octanal in hazelnut flour and determined that the amount of hexanal was high in the samples [89]. The author explained this situation by noting that hexanal is formed as a result of oxidation of linoleic acid, octanal is formed as a result of oxidation of oleic acid, and linoleic acid is oxidized faster than oleic acid [89]. n-Alcohols and fatty acids were also determined in the air space of HOC. These compounds were identified in hazelnut samples in previous studies [90].

Benzaldehyde, an aromatic aldehyde and a metabolite of phenylalanine, constituted 48.8% of the headspace of FHOC samples. This compound might be produced during fermentation because Tabanelli et al. (2018) determined the benzaldehyde increment with fermentation in hazelnut-based food [91]. In addition to these compounds, the formation of 3-methylbutanal, 3-methylbutanoic acid, and 2-methylpropanoic acid was determined in FHOC. These compounds are formed by catabolism of branched chain amino acids [92]. Another abundant compound in the headspace was 2-nonanol (17.6%), which is a methyl ketone group. Methyl ketones (2-alkanones) are derived from medium length fatty acids [93]. 2-heptanone from the methyl ketone family was also detected in the headspace of the FHOC. Another group of compounds detected in fermented samples were furans, pyrazines and pyrroles. These compounds are the dominant compounds in heat-treated foods. There are different pathways responsible for furan formation which are based on pyrolysis of sugars at high temperatures, decomposition to ascorbic acid and related compounds, Maillard-type reaction systems involving amino acids and reducing sugars, and oxidation of polyunsaturated fatty acids [94]. The formation of pyrazines and pyrroles is also associated with Maillard reactions, and it is known that their formation is associated with an increase in the amount of free amino acids [95]. Finally, another dominant compound in the headspace is 1-octen-3-ol, a compound frequently detected in fermented foods that can be produced by microorganisms through the metabolism of unsaturated fatty acids [96].

### 2.6. Bioactive Properties

The fermentation process increased the number of soluble phenolic compounds in HOC by 3.5 times and the total number of soluble peptides by 14.3 times (Table 7). The highest amounts of phenolic compounds were determined in aqueous extracts of HOC and FHOC, followed by 80% methanolic extract and methanolic extract. The fermentation process increased the amount of phenolics dissolved in methanol, which corresponds to a 48-fold increase. The process also increased the phenolic contents in 80% methanolic extract and aqueous extract but the increments were more limited as 7-fold and 3.5-fold, respectively. Similar increments in phenolic content during fermentation have been reported in the literature. The amounts of water-soluble and methanol-soluble phenolic compounds in rice samples fermented by *Aspergillus oryzae* increased 8% and 244%, respectively, compared to the non-fermented rice [97]. Similarly, the phenolic compound contents of fermented wheat, brown rice, maize, and oats increased significantly [98]. There are many studies showings that the fermentation process converts conjugated phenolic compounds into free phenolic compounds by microbial-derived enzymatic activities [99]. It is thought that the enzyme β-glucosidase (beta-D-glucoside glucohydrolase, (E.C. 3.2.1.21) is particularly effective in the increase of phenolic compounds. This enzyme hydrolyzes phenolic glycosides to form free aglycones with high antioxidant activity [100]. Extracellular enzymes produced by *Aspergillus oryzae* include α-amylase, glucoamylase, α-glucosidase, cellulase, β-galactosidase, polygalacturonase, pectin lyase, and xylanase [49,101]. These enzymes are especially effective in the release of phenolic compounds in hazelnut meal, which are bound to carbohydrates, into the environment in free form. Lignocellulolytic enzymes also break down polysaccharides in the cell wall, hydrolyze insoluble cellular components and release bioactive components such as phenolic acids bound to the cell wall. In addition, the activities of β-glucosidases, which hydrolyze phenolic glycosidases, and phenolic esterases, which can liberate phenolic compounds upon disruption of cell integrity, may also have contributed to the increase in total phenolics in fermented hazelnut meal [102].

At the end of the fermentation process, changes in the bioactive properties of hazelnut meal were expected as a result of various hydrolysis reactions occurring with the activities of microbial enzymes in the medium. The emergence of free amino acids and bioactive peptides as a result of protein hydrolysis due to the high extracellular enzymatic activities of *Aspergillus oryzae*, the increase in free phenolic compounds and water-soluble carbohydrate content as a result of alpha amylase, and other glucosidase enzyme activities may be the main factors that cause significant changes in the bioactivity of fermented hazelnut meal [82,103]. The bioactive properties of food products, such as antioxidant, antimicrobial, antihypertensive, antidiabetic, etc., are generally associated with their phenolic components and bioactive peptides.

On the other hand, there was a limited increase in the amount of total soluble protein content, from 256.8 ± 3.3 to 279.5 ± 4.0 mg BSA/g. Teng et al. (2012) reported an increase in the soluble protein content of soybean meal fermented with *Aspergillus oryzae*, but the increase was limited, close to 20% [82]. However, the results showed that there was a high increase in the number of primary amino groups in the medium during the fermentation process which was expressed as a total peptide content (TPC). The method used to determine the TPC is based on the determination of the compound formed as a result of the reaction between primary -NH_2_ groups and OPA (ortho-phthalaldehyde) in the presence of reduced sulfhydryl groups by spectrophotometric methods. The main primary amino groups in the medium are located at the N-terminal end of the polypeptide chain and at the epsilon amino group of the amino acid lysine. Hydrolysis of the polypeptide chain increases the number of amino groups in the medium. The main reasons for this increase may be the proteolytic activities that occur as a result of microbial activities (proteases hydrolyze polypeptide chains and increase the amount of free amino acids and peptides with primary amino groups), because *Aspergillus oryzae* has the ability to produce proteases such as extracellular alkaline protease, aspartic protease, neutral protease II, carboxypeptidase [49]. Although there is an increase in the amount of lysine amino acid in the medium after fermentation (hazelnut meal lysine 1.64%, fermented hazelnut meal lysine 2.01%), it is thought that the main effect on the result is more likely to be caused by the primary amino groups of free amino acids and peptides released as a result of protein hydrolysis.

In this study, it was also observed that there were significant increases in the antioxidant activity values of the FHOC during the fermentation process. The fermentation process increased the ABTS cation radical scavenging activity of the meal by 3.8 times and DPPH radical scavenging activity by 4.9 times. Although there was a significant increase in hydroxyl radical scavenging capacity, this increase was quite limited compared to other antioxidant activities. In the literature, it was reported that there were significant increases in the antioxidant activities of the end products formed when different plant sources (wheat, oat, buckwheat, soybean, chickpea, rice bran, cocoa) were fermented with solid or liquid fermentation methods using various microorganisms (bacteria, yeast, mold) [99]. Studies have shown that the DPPH radical scavenging activity of soybean meal fermented by SSF method using *Aspergillus oryzae* increased 2.5 to 5.5 times after fermentation [51,104]. Similarly, this activity increased approximately 3-fold with rice fermentation [97]. Bhanja Dey and Kuhad (2014) reported significant increases in DPPH radical and ABTS^+^ radical scavenging activities of fermented wheat, brown rice, corn, and oats in their study [98]. After fermentation of flaxseed meal by solid culture fermentation using *Aspergillus oryzae*, the ABTS cation radical scavenging activity of the meal increased depending on the fermentation time [80]. In a study in which hazelnut milk was fermented with kefir grains, the DPPH radical scavenging activity of the beverage increased (60%) after fermentation [105]. Although this increase is quite low compared to the increase in our study, it is thought that the type of fermentation (dry, liquid), type of microorganism (bacteria, mold), and type of material (hazelnut meal, hazelnut kernel) have a decisive impact on the result. Maleki et al. (2015), while preparing the hazelnut samples to be fermented, separated the dark colored and phenolic-rich membrane surrounding the hazelnut kernel [105]. This process is estimated to significantly reduce the phenolic content of the raw material. As Shahidi et al. (2007) stated, the highest phenolic compound content is in the membrane covering the hazelnut kernel [106]. In terms of content, the membrane is followed by the brown hard shell outside the hazelnut (about 50% of the phenolic component of the membrane). In parallel with the phenolic content, the membrane also showed the highest antioxidant activity. Antioxidants such as ascorbic acid, tocopherol, and polyphenols can stabilize radicals by giving electrons or hydrogen atoms to free radicals from their hydroxyl groups. In addition, they can prevent the formation of free radicals through various reactions by chelating ions such as Fe^2+^, Fe^3+^, Cu^2+^, and Cu^+^ in the environment [107]. Similarly, proteins, especially peptides, can show antioxidant activity through the various mechanisms. Studies have shown that bioactive peptides with antioxidant activity are generally composed of 3–6 amino acid units and have molecular weights less than 1 kDa. Approximately 1/3 of these antioxidant peptides are composed of glycine, proline and leucine amino acids. In addition to these, it was determined that histidine amino acid with imidazole ring showed strong antioxidant activity due to its ability to give proton. Tryptophan and proline amino acids with indole and pyrrolidine groups can function as hydrogen donors through their hydroxyl groups. Amino acids with excess electrons, such as glutamic acid and aspartic acid, also have the ability to bind free radicals. In addition to these properties, the amino acid sequence of the peptide has also been reported to have a significant effect on antioxidant activity [108,109].

There is an increasing interest in the inhibitors developed or discovered for enzymes that catalyze certain reactions in metabolism and cause undesirable conditions in the body. In this study, the inhibition activities of α-amylase, α-glucosidase, angiotensin converting enzyme (ACE), and acetylcholine esterase (AChE) of hazelnut meal were determined by fermentation process. In general, α-amylase and α-glucosidase enzymes are associated with weight gain, calorie intake, and glycemic index values as they are responsible for the digestion of carbohydrates [110]. ACE is an important enzyme that catalyzes one step of a series of biochemical reactions that can lead to high blood pressure, and ACE inhibitors such as captopril are used in the treatment of high blood pressure illness [2]. AChE inhibitors are also used in the treatment of neurodegenerative diseases such as Alzheimer’s disease [111]. However, the in vitro analyses performed here only reflect the enzyme inhibitory potential of HOC and FHOC. Under in vivo conditions, these inhibition effects may be different from the results determined here. In the present study, in accordance with the literature, α-amylase inhibition was not observed in FHOC (10 mg/mL, pH 7.0), but the FHOC itself had enzyme activity of α-amylase (17.52 ± 0.55 U/mL).

The α-glucosidase enzyme inhibition value of FHOC increased from 10.49 ± 0.12 to 182.11 ± 2.61 mg acarbose/g by fermentation. A very high inhibition increase was observed here (*p* ≤ 0.05). Although *Aspergillus oryzae* produces various glucosidase enzymes to hydrolyze complex or polymeric carbohydrates, a study showed that out of 13 different *Aspergillus oryzae* strains, one extracellular and ten intracellular strains had α-glucosidase inhibitory activity [112]. When one of the strains was examined in detail, it was determined that the inhibitory activity increased as the fermentation time increased. As a result of the analysis, it was reported that Cys-Leu and Pro-Phe-Pro peptides showed very high inhibitory effect. In fermented lentil samples produced by SSF using *Aspergillus oryzae* and *Aspergillus niger*, the α-glucosidase inhibition activity decreased over time in the samples in which *Aspergillus oryzae* was used during fermentation, while the inhibition activity increased over time in the medium in which *Aspergillus niger* was used. In that study, the inhibitory effect was attributed to the presence of bioactive peptides in the medium [20]. In another study, the increased α-glucosidase inhibition activity in mulberry fermentation using *Aspergillus oryzae, Aspergillus niger*, and *Monascus anka* was associated with aglycone phenolic compounds (quercetin and kaempferol) [113]. A similar approach was reported for the increased inhibitory effect of black tea fermentation [114].

A 43% decrease in ACE inhibition activity of FHOC was observed after fermentation. ACE inhibition activity is generally associated with the binding of biomolecules such as polyphenols, flavonoids and bioactive peptides to the active site of the enzyme, thereby reducing enzymatic activity [115]. Most studies in this field have been carried out with bioactive peptides produced from milk proteins, and the results have shown that short-chain peptides are most likely to exhibit high antihypertensive activity if they have proline, lysine, or arginine amino acids at their C-terminal ends. For long-chain peptides, it has been reported that the last four amino acids at the C-terminus are decisive in antihypertensive effect [116]. They reported significant increases in the amount and activity of antihypertensive peptides in soybeans fermented using *Aspergillus oryzae* [117]. Similar to our study, in 12 buckwheat flours fermented using 13 different lactic acid bacteria, decreases in ACE inhibition activities were observed in varying ratios compared to the control sample [118]. Puspitojati et al. (2019) reported that during tempeh production from legumes, ACE inhibition activity in the medium increased until the 3rd day of fermentation, then decreased sharply [119]. Similar to our study, as the proteolysis time or activity increases, the amino acids in the C-terminal of antihypertensive peptides can be hydrolyzed and become free, and the antihypertensive peptide may lose its activity. Rezaei et al. (2019) reported that after a certain time of fermentation during yogurt production, microorganisms in the environment hydrolyze and consume ACE inhibitory peptides formed in the early stages of fermentation [120]. Considering the studies in the literature, it can be considered that the fermentation period is long enough for the production and subsequent hydrolysis or consumption of ACE inhibitory peptides.

### 2.7. Functional Properties of HOC and FHOC

The WAC and OAC of HOC were measured as 5.31 ± 0.06 and 2.14 ± 0.06 (g/g), respectively (Table 8). These values are slightly different than the values Tatar et al. (2015) reported in the literature [121]. They measured the WAC of HOC as 4.10 ± 073 g/g, which is slightly lower, whereas the OAC (3.38 ± 0.11 g/g) was slightly higher than our results. These slight differences might be caused by the different de-oiling process of hazelnut flours or type of the hazelnuts used in the studies. In general, the WAC seems higher than the OAC of hazelnut cakes or flours. The higher WAC of de-oiled hazelnut cake might be attributed to its protein and dietary fiber content, which have many functional groups tending to interact with water molecules [61,121]. It has been reported that the WACs of cake/meal such as soybean, sunflower, canola, flaxseed, hemp, milk thistle, poppy, pumpkin, rapeseed, and safflower are lower than the WAC of the HOC [59,62,122,123]. On the other hand, the WAC of flaxseed cake was found to be higher than HOC by Khattab and Arntfield (2009) [62]. Similar to these results, the OAC of HOC was measured higher than those of soybean, sunflower, flaxseed, hemp, milk thistle, poppy, pumpkin, rapeseed, and safflower cake in the literature [59,122,123]. In addition to this, Khattab and Arntfield (2021) reported that the OACs of soybean, canola, and flaxseed meals used in their study were similar to the OHC of HOC used in our study [62].

The fermentation process led to remarkable changes in the functional properties of HOC. While WAC decreased after fermentation, significant increases in OAC were measured. After the fermentation process, the WAC of HOC decreased by 52.3%. Oloyede et al. (2016) reported for moringa flour that while there was a steady increase in the WHC of the flour during the first 48 h of fermentation, a statistical decrease in the WHC value started at 72 h of fermentation [124]. This can be explained by the fact that in the first stages of fermentation, the water absorption ability of some polymeric carbohydrates increases due to the release of more functional groups with partial hydrolysis, and as a result of progressive fermentation, partially hydrolyzed polymeric structures are further hydrolyzed into monomers or oligomers and lose their water absorption ability by dissolving in water [124]. The fermentation time applied in our study may have caused FHOC to have a lower WAC value for this reason. On the other hand, the OAC value of FHOC was 55% higher than that of HOC by fermentation. A similar result was observed in the OAC values of soybean cakes fermented using *L. plantarum*, *L. brevis*, and *L. acidophilus*. Significant increases in the OAC of the sample were detected depending on the type of microorganism used after fermentation [125]. Oloyede et al. (2016) similarly reported that the OAC of moringa flour increased with fermentation time [124]. The higher OAC of FHOC can be attributed to the fact that more hydrophobic protein groups are exposed after hydrolysis and absorb oil molecules and the OAC increases with the help of more hydrophobic interactions as a result of decreased protein solubility by decreasing medium pH [62,125]. As the medium pH approaches the isoelectric point of proteins in FHOC, the surface charge and solubility of proteins begin to decrease. As a result, hydrophobic interactions increase, allowing proteins to interact with more oil molecules [2,125].

An emulsion consists of two immiscible liquid systems where one of the liquids is dispersed as small spherical droplets into another liquid, formed by mechanical energy [126]. For a formation of a homogenic emulsion system, a small concentration of surfactant which is a compound has the ability to bind both hydrophilic and hydrophobic groups is needed. The stable emulsion systems are required for food formulations; therefore, the emulsion formation ability is one of the important technologic factors of food components as flours, cakes, and protein to assess their usability in food production. In this study, the emulsion activity index (EAI) was measured for HOC and FHOC and the values were 42.32 ± 2.18 and 36.74 ± 1.42 m^2^/g, respectively. However, the fermentation process was not considerably affecting the EAI of FHOC, the emulsion stability of HOC and FHOC was significantly different from each other. The emulsion stability index (ESI) of HOC was 175 ± 10 min after 60 min incubation at 25 °C while ESI of FHOC was 763 ± 32 min (Figure 2). This ESI of HOC was considerably higher than the ESI of emulsion formed by defatted hazelnut cake in another study conducted by Tatar et al. (2015) [121]. They found the ESI value of their sample as 31.9 ± 2.43% after ten minutes incubation while ESI of our sample was 31.70 ± 4.47% after 15 min incubation. Singh and Koksel (2021) reported that the 33 emulsions produced by different soybean meal had ESI values varied between 59.30 ± 1.16 and 79.51± 1.6% after 30 min incubation where the ESI was 81.26 ± 6.86% for the emulsion by HOC [123]. On the other hand, Rani and Badwaik (2021) determined increased ESI stability more than 100% for mustard, flaxseed, and soybean cake when the emulsions were heated at 80 °C for 30 min [61]. The similar method was applied by Petraru et al. (2021), who determined the emulsion stability of sample produced by sunflower oil cake was 27.92 ± 0.57% [122]. The remarkable emulsion stability in FHOC might be attributed to the exposed hydrophobic groups or partially hydrolyzed proteins by metabolic activity of the microorganism. Moreover, more protein compounds were also produced during fermentation which might be contribute to emulsion stability of FHOC. The emulsion stability of the cakes was monitored for 1440 min and the stability measurements were also determined at 15th, 30th, 45th, 60th, 180th, and 1440th minutes of incubation. The decrease in emulsion stability of HOC was dramatic by time while the slight stability decrease was observed in emulsion formed by FHOC. After 1440 min (24 h) of incubation, 37.72% of stability emulsion formed by HOC was lost. In contrast, the emulsion formed by FHOC only lost 71.6% of its stability.

Foam is defined as a system where the air is dispersed in a liquid and separated by a thin continuous liquid layer [121]. The foaming capacity (FC) is expressed as the formed foam volume in the tube by mechanical energy. The FC of HOC was 10.96 ± 1.41 mL, which was almost two times more than the FC of hazelnut oil cake prepared at pH 9.5 and 5.0 by Tatar et al. (2015) [121].

## 3. Materials and Methods

### 3.1. Solid Substrate and Chemical Materials

Hazelnut oil cake, as a by-product of cold-press oil production, was supplied by Cansızzade Doğal Bitkisel Yağlar Company (Istanbul, Türkiye). After removing the foreign materials, the cake was ground by a centrifugal mill (Retsch ZM200, Haan, Germany) into powder with 250 µm average particle size. The prepared powder was kept in polyethylene bags at 4 °C until fermentation. Potato Dextrose Agar and Tween 80 were purchased from Merck, Darmstadt Germany. o-phthaldialdehyde, di-Na-tetraborate decahydrate, Na dodecyl-sulfate, Potassium-sodium tartrate, Folin–Ciocalteu phenol reagent, Ferrozine^®^, copper (II) sulphate, iron (II) chloride tetrahydrate, angiotensin-converting enzyme, captopril, N-[3-(2-Furyl)acryloyl]-Phe-Gly-Gly (FAPGG), α-amylase from porcine pancreas, α-glucosidase from *Saccharomyces cerevisiae*, 3,5-Dinitrosalicylic acid (DNS), starch, 4-nitrophenyl α-D-glucopyranoside, subtilisin A-Alcalase^®^ (4860), Savinase^®^ (P311), 2,2-diphenyl-1-picrylhydrazyl (DPPH), 2,2′-azinobis-(3-ethylbenzothiazoline-6-sulfonate) (ABTS), 2,4,6-tris(2-pyridyl)-Striazine (TPTZ), gallic acid, trolox, 4-(2-aminoethyl) benzenesulfonyl fluoride hydrochloride (AEBSF), DTNB (5,5′-dithio-2-nitrobenzoic acid), and acetylcholinesterase, acetylthiocholine iodide were purchased from Sigma-Aldrich Co., St. Louis, MO, USA. Other chemicals, such as NaOH, HCl, NaCl, etc., were used at analytical grade.

### 3.2. Microorganism and Inoculum Preparation

*Aspergillus oryzae* strain 200828, which was identified for its amylase, protease, and lipase production ability, was purchased from the Fungal Culture Collection of TÜBİTAK Marmara Research Center and kept in glycerol stock at −80 °C. *A. oryzae* was propagated on Potato Dextrose Agar (PDA) plates at 30 °C for 6 days. The spores on PDA plates were harvested by adding 10 mL of sterile Tween 80 (0.01% *v*/*v*). The spore suspension was collected in a sterile falcon tube and stored at 4 °C until the inoculation step within 4 days. The initial spore counts were determined using a hemocytometer (Isolab, Adana, Türkiye) just before inoculation process.

### 3.3. Solid State Fermentation

Solid-state fermentation was conducted adding appropriate amount of substrate (Table 9) into 250 mL Erlenmeyer flasks according to Demir and Tari (2014) [63]. The amount of liquid to maintain the desired initial moisture content was calculated. Half of the required water was added to the medium before sterilization (121 °C, 15 min). The other half of the water (sterilized) was used for the homogenous spread of inoculum into solid substrate. The prepared and inoculated flasks were fermented in a static incubator (Memmert IN 110, Schwabach, Germany) under the conditions detailed in Table 9. Two replicates were prepared and fermented for each run of the experimental design. Amount of solid substrate, inoculum ratio, and particle size were kept at 15 g, 10^7^ spore/g DM substrate, and 500 µm levels, respectively. The FHOC was kept in a refrigerator for further extraction and analysis.

### 3.4. Experimental Design and Statistical Analysis

A Box–Behnken experimental design (BBD) with 17 runs (5 center points) and RSM were employed to investigate the effects of initial moisture content, incubation temperature, and incubation time on the total peptide content of HOC and to determine the optimum conditions maximizing TBC [67,127]. The lower (−1) and upper (+1) levels of SSF conditions were selected based on preliminary experiments. An average of two replicates of total peptide content (mg tryptone eqv./g DM of solid substrate) was taken as the response variable.

The results of the conducted experiments were used to express the relationship between the predicted response and the process parameters by a second-order polynomial regression model (Equation (2)).
(2)$$Y=\beta_{0}+\overset{k}{\underset{i=1}{\sum }}\beta_{i}X_{i}+\overset{k}{\underset{i=1}{\sum }}\beta_{\mathrm{ii}}X_{i}^{2}+\overset{k}{\underset{{İ}_{İ=1}}{\sum }}\overset{k}{\underset{j>1}{\sum }}\beta_{\mathrm{ij}}X_{i}X_{j}$$

In Equation (1), *Y* represents the predicted response, *β*_0_ is the constant term, *β*_i_ is the linear coefficient, *β*_ii_ is the quadratic coefficient, and *β*_ij_ is the interaction coefficient. The experiments were performed, and variance analysis was statistically conducted using Design Expert (11.0.0 trial version, Stat-Ease, Minneapolis, MN, USA) software. The significance of the model was evaluated according to ANOVA, lack-of-fit test, and multiple determination coefficients tests. In the regression analysis, the variables having a *p*-value lower than 0.05 were accepted to have significant effect on the TPC.

After the model compatibility was ensured, the validation of the assumptions used in ANOVA was performed with statistical calculations. In the optimization, the “desirability function” method, which is based on the simultaneous solution of the models determined for each response, was used. Experimental validation of the determined optimum conditions was performed by the production and analysis under the specified conditions. The criteria for the solution were “in range” between the upper and lower limits of the investigated factors and “maximize” for the response. *Error* (%) between the experimental (*y_exp_*) and predicted (*y_pre_*) values was calculated by Equation (3).
(3)$$Error\;\left(\%\right)=\frac{\left|y_{exp}-y_{pre}\right|}{y_{exp}}\times 100$$

### 3.5. Extraction of Bioactive Materials from SSF Media

After the SSF is terminated, the fermentation media was frozen at −20 °C (Arçelik, İstanbul, Türkiye) immediately. Subsequently, the frozen samples were lyophilized, dried, and pulverized (LABART LFD-10N, Gdansk, Poland). The dried samples were stored in zip lock polyethylene bags at −20 °C until analysis (Torino et al., 2013) [33]. For bioactivity analysis, a 10 g sample from the culture medium was extracted by 100 mL distilled water in a shaking incubator (IKA KS3001, Staufen, Germany) 100 rpm, for 1 h at room temperature, followed by centrifugation (8000 rpm, 4 °C, 20 min, Universal 320R, Hettich, Co., Tuttlingen, Germany). The supernatant was used for analysis [98].

### 3.6. Determination of Total Peptide Content (TPC)

TPC was determined spectrophotometrically following OPA (o-phthaldialdehyde) derivatization [111]. On 0.1 mL of 4% (mg/mL methanol) OPA, 2.5 mL of 100 mM sodium tetrahydroborate solution, 0.25 mL of 20% sodium dodecyl sulfate solution, and 10 µL of β-mercaptoethanol solution were added. The total volume of this daily prepared solution was up to 5 mL with water. A volume of 10 µL of the hydrolysate solution was mixed with 200 µL of OPA solution and incubated for 2 min at room temperature. The blank solution was similarly prepared with water instead of hydrolysate. Tryptone was used as the standard solution. Solution absorbances were determined spectrophotometrically at 340 nm and expressed as mg tryptone equivalents/g dry matter of solid substrate.

### 3.7. General Characterization of HOC and FHOC

Total moisture content was determined gravimetrically, total protein content was determined by the Kjeldahl method with a protein conversion factor of 6.25 (Gerhardt Vapodest, Königswinter, Germany), soluble protein content was determined by the spectrophotometric method (Agilent Cary 60 UV-Vis, Agilent Technologies, Santa Clara, CA, USA), total fat content (Gerhardt, Soxtherm SE414, Königswinter, Germany), and total ash content was determined gravimetically according to methods given in Aydemir et al. (2014) and Aydemir et al. (2022) [2,110]. Total carbohydrate content was obtained by calculating the total amount of protein, fat, moisture, and ash in a 100 g sample and subtracting it from 100. Color analysis of the products was performed with a color meter (Konica Minolta CM-5, Osaka, Japan) and the results were given as L*, a*, and b*. Total phenolic content was determined using Folin–Ciocalteu and the results were expressed as gallic acid equivalents [110]. Extractions from FHOC for phenolic matter determination were carried out using distilled water, methanol, and a mixture of methanol:water (80:20 by volume) as solvents. In the extraction process, 2 g of sample was extracted in 10 mL of solvent in a shaking incubator (IKA KS3000 IC, Staufen, Germany) for 24 h at room temperature and the supernatant obtained after centrifugation (7500 rpm, 25 °C, 30 min) was used for the determination of phenolics. All compositional analyses were performed in triplicate and the results were expressed as dry weight.

### 3.8. Determination of Amino Acid Composition of HOC and FHOC

Sample hydrolyzation: Acid and basic hydrolysis were applied to determine the amino acids in the protein structure. Acid hydrolysis was applied for amino acids other than tryptophan. For acidic hydrolysis, the samples were weighed considering their protein content (1 mL of 6 M HCl hydrolyzes 4 mg protein) and 6 M HCl containing 0.1% phenol was added to them; the samples were kept under nitrogen gas and left to hydrolysis in an oven at 110 °C for 24 h. After extraction, the pH of the cooled hydrolyzed samples was adjusted to 2. Basic hydrolysis was performed for tryptophan because it is affected by acidic hydrolysis. Weighed samples were subjected to hydrolysis at 110 °C for 24 h by adding 4 M LiOH and 50 mg ascorbic acid. The pH of the cooled hydrolyzed samples was brought to 6–7. The hydrolyzate samples were passed through a 0.45 µm filter (PVDF) and stored for derivatization.

Before derivatization, 10 mL of the hydrolysates were taken and 800 µL of norvaline internal standard at a concentration of 250 mg/L was added and the volume was completed to 20 mL.

Pre-column derivatization with OPA: Derivatization with OPA was carried out using the method of Wang et al. (2010) with some modifications [128]. OPA solution for derivatization was prepared daily. First, 10 mg o-phthalaldehyde was dissolved in 0.1 mL methanol. Then, 0.9 mL boric acid buffer (0.4 M, pH 9.0) and 0.5 mL mercaptoethanol were taken in an amber vial, mixed, and the OPA solution was prepared.

HPLC-DAD analysis: Determination of amino acids was performed by HPLC-DAD system (Shimadzu, Kyoto, Japan). A volume of 1000 µL of the prepared samples were taken, and 600 µL of OPA solution was added and mixed with the help of vortex for 10 s. After incubation at room temperature for 5 min in the dark, it was injected into the HPLC column. Separation of the derivatized amino acids was carried out on a reverse phase Zorbax Eclipse-C18 column (150 × 4.6 mm, 5 μm, Agilent Technologies, Santa Clara, CA, USA) at a column temperature of 40 °C and a solvent flow rate of 1.4 mL/min. The injection volume was set to 10 µL. Phosphate buffer (0.04 M, pH 7.2) was used as mobile phase A and methanol:acetonitrile:phosphate buffer (45:45:10, *v*:*v*:*v*) was used as mobile phase B. Elution was performed using a linear graded elution program (Table 10). Derivatized amino acids were detected at 338 nm with a DAD detector. To determine the amount of free amino acids in the samples, standard amino acid mixtures (prepared with 0.01% HCl) were injected into the HPLC under the same conditions. Quantification was performed on the calibration graphs generated. The internal standard (L-norvaline) technique was used for quantification. Proline and cysteine amino acids could not be determined by using OPA derivatization method with the HPLC-DAD system. Asparagine, glutamine, and methionine amino acids were not visible in the chromatogram because they were affected by acid hydrolysis. Therefore, quantification was not possible [128].

### 3.9. Determination of Volatile Compounds of HOC and FHOC

Identification and quantification of volatile compounds were carried out by Gas Chromatography-Flame Ionization Detector (GC-FID) (Agilent 6890N, Agilent Technologies, Santa Clara, CA, USA) and Mass Spectrometry (MS) (Agilent 5975B VL MSD, Agilent Technologies, Santa Clara, CA, USA). Solid Phase Micro-Extraction (SPME) technique was used for the isolation of volatile compounds. Equilibration, extraction, and injection were performed with the help of an automatic injection module (GC Injector 80, Agilent Technologies, Santa Clara, CA, USA). Mixing on/off time was 5/2 s, vial needle penetration was 11 mm, and vial fiber open zone was 22 mm. Divinylbenzene Carboxene/Polydimethylsiloxane (DVB/CAR/PDMS, 50/30 μm, SF 23GA Auto, 57299-U, Sigma-Aldrich Co., St. Louis, MO, USA) coated fiber was used for the isolation of volatile compounds. The extraction conditions considered for the samples to be analyzed were those determined in Salum et al. (2017) (54.8 °C extraction temperature, 86 min incubation time, and 250 rpm mixing speed [129]). Fiber desorption took place at 250 °C for 180 s at the GC injection port (7890B, GC System, Agilent Technologies, Santa Clara, CA, USA). Separation was performed on a DB-Wax (30 m × 250 μm × 0.25 μm; 122-7032, Agilent Technologies, Santa Clara, CA, USA) polar column, using helium gas as carrier gas at a flow rate of 2 mL/min. After 2 min at 40 °C, the oven temperature was increased by 5 °C per minute to 70 °C, held for 1 min, then increased by 10 °C per minute to 240 °C and programmed to remain constant at this temperature for 4 min. The column current was then transferred to the FID and MS detectors. The detector temperature was set to 260 °C. The MS ionization energy was 70 ev and scanned between 30–400 mass/charge. The internal standard method was used to quantify the compounds. Deconvolution of the chromatograms was performed with AMDIS (Automated Mass Spectral Deconvolution and Identification Software, version number 2.70). MS libraries (NIST 11, Wiley 7.0) and standard compounds were used for peak identification (Salum et al., 2022) [126]. In addition, alkane series (C8–C20) were injected and linear retention indices (LRI) of the peaks were calculated.

### 3.10. Analysis of Bioactive Properties of HOC and FHOC

Determination of ABTS^+^, DPPH, and hydroxyl free radical binding capacity: The free radical binding capacities of the samples were determined spectrophotometrically [130]. The reaction of 0.1 mL of sample solution with 1.9 mL of 7 mmol/L ABTS radical solution (prepared in 2.5 mmol/L potassium persulfate) was determined kinetically by measuring the change in absorbance at 734 nm for 6 min. For hydroxyl radical binding capacity, 100 μL of sample and 3 mL of Smirnoff reagent were incubated at 37 °C for 30 min. Absorbance measurements were then performed at 510 nm. The results were expressed as trolox equivalents.

Determination of iron chelating activity: The iron ion chelating capacity of the samples was determined by spectrophotometric method [130]. A volume of 2 mL of appropriately diluted sample solution was mixed with 0.1 mL of 1 mmol/L FeCl_2_·4H_2_O solution and incubated for 30 min. Then, 0.1 mL of 5 mmol/L ferrozine was added, mixed, and incubated for another 10 min. The resulting color was measured spectrophotometrically at 562 nm and the results were expressed as EDTA conjugate.

Determination of Angiotensin Converting Enzyme (ACE) inhibition activity: A volume of 10 μL of 0.25 units/mL ACE prepared in 0.01 mol/L saline phosphate buffer (pH: 7.0, NaCl concentration: 0.5 mol/L) was mixed with 10 μL sample solution. The mixture was incubated at 37 °C for 15 min, then the reaction was initiated by adding 150 μL of 1.75 mmol/L FAPGG substrate solution (37 °C) prepared in saline phosphate buffer. The absorbance of the reaction was kinetically measured at 340 nm at 37 °C for 30 min. ACE activity was determined using the slope of the initial linear region of the absorbance–time curve and captopril was used as control [2].

Determination of α-amylase and α-glucosidase inhibition activity: α-glucosidase and α-amylase inhibition assays were performed according to the methods described in Aydemir et al. (2022) [110]. A volume of 100 μL of α-glucosidase enzyme (1 unit/mL) prepared in 0.1 mol/L sodium phosphate buffer (pH 6.8) was pre-incubated with 50 μL sample at 37 °C for 10 min. Then, the enzymatic reaction was initiated by adding 50 μL of 10 μM (4-Nitrophenyl β-D-Galactopyranoside) as substrate and the reaction at 37 °C for 30 min was terminated by adding 1 mL of 0.1 mol/L Na_2_CO_3_. Enzyme activity was determined by measuring the p-nitrophenyl formed at 400 nm. For the other assay, 100 μL of α-amylase enzyme (1 unit/mL) prepared in 0.1 mol/L sodium phosphate buffer (pH 6.8) was incubated with 100 μL of sample for 5 min at 37 °C and 250 μL of 1% (mg/mL) starch prepared in sodium phosphate buffer (pH 6.8) was added to initiate the reaction. The reaction was carried out at 37 °C for 5 min and 200 μL of DNS reagent (1% 3,5-dinitrosalicylic acid and 12% sodium potassium tartrate in 0.4 mol/L NaOH) was added. The reaction was carried out by heating at 95 °C for 10 min. It was then diluted with 1 mL of distilled water in an ice bath. α-amylase activity was determined by measuring absorbance at 540 nm. Sample-free solution as control, substrate-free solution as blind, and enzyme-free solution as enzyme-blind were used (for each sample).

Determination of acetylcholine esterase (AChE) inhibition activity: The inhibition activity of the samples was carried out as described in Vinutha et al. (2007) [131]. The principle of inhibition activity is the measurement of the yellow anion TNB (5-thio-2-nitrobenzoate), which was released as a result of the reaction between thiocholine and DTNB (5,5′-dithio-2-nitrobenzoic acid) hydrolyzed by acetylcholinesterase, at 412 nm absorbance acetylthiocholine iodide (10.85 mg/5 mL phosphate buffer) was used as substrate for activity determination. The solution containing various concentrations of sample or standard to be prepared for analysis was prepared with 250 µL of 200 mM phosphate buffer at pH 7.7, 80 µL DTNB, and 10 µL enzyme (2 U/mL). The solution was incubated at 25 °C for 5 min for pre-incubation. Then, 15 µL of substrate was added to the solution and incubated again for 5 min, and the yellow-colored compound was formed.

### 3.11. Analysis of Functional Properties of HOC and FHOC

The functional properties of HOC and FHOC were determined according to the methods described in Aydemir and Yemenicioglu (2013) [58].

## 4. Conclusions

The solid-state fermentation process was optimized for hazelnut oil cake by using *Aspergillus oryzae* and the optimum conditions were determined as initial moisture content of 69.8%, incubation temperature of 24.6 °C, and incubation time as 4.6 days. The fermented hazelnut oil cake had higher protein, oil, and ash content, but lower carbohydrate content, than unfermented hazelnut oil cake. About 6.1% more essential amino acids were produced during fermentation and 48.8% was benzaldehyde among the determined volatile compounds in the fermented product. During fermentation, the peptide content was fourteen times (462.37 mg tryptone/g) and the phenolic content was seven times (11.30 ± 0.39 mg GA/g) greater with higher ABTS+ (75.61 µmol Trolox/g), DPPH (14.09 µmol Trolox/g), OH (265 mg ascorbic acid/g) radical scavenging, and α-glucosidase inhibition activity (182.11 ± 2.61 mg acarbose/g) in fermented hazelnut oil cake. The increment in peptide and free soluble phenolics might be associated with microbial activities, especially enzymatic activities of digestive enzymes such as proteases and glucosidases that hydrolyzes macromolecules as proteins and carbohydrates. Although the significant improvements were determined in fermented hazelnut oil cake, the fermentation process provides a limited contribution to functional properties of the cake. The solid-state fermentation carried out by Aspergillus oryzae revealed that it is a green and economic way to improve the bioactive component of the hazelnut oil cake, which has the potential to be used as a source of bioactive food additive in food formulations. However, different types of food formulation should be studied to see the actual effect of importing bioactivity to the processed food in order to understand the suitability of usage fermented hazelnut oil cake. On the other hand, the main limitation of this study in terms of industrial applications is being a lab-scale study. Pilot-scale and industrial-scale studies should be conducted to optimize the fermentation conditions because the change in the scale may significantly affect the optimum process conditions.

## Figures and Tables

**Figure 1 molecules-29-04237-f001:**
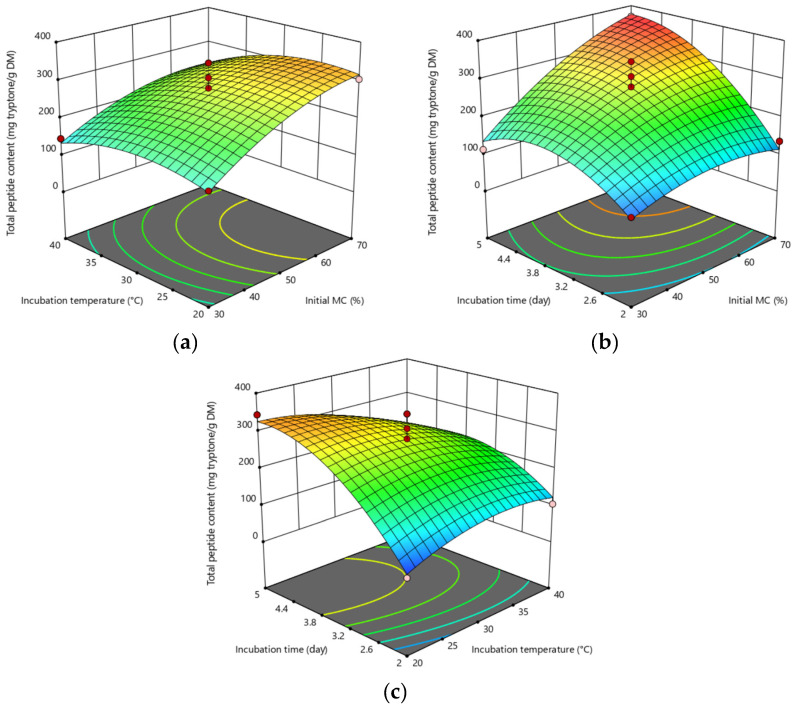
Response surface plots for the effects of (**a**) initial moisture content–incubation temperature (incubation time hold at 3.5 days), (**b**) initial moisture content–incubation time (incubation temperature hold at 30 °C), and (**c**) incubation temperature–incubation time (initial moisture content hold at 50%) interactions on TPC. (minimum TPC region: blue, maximum TPC region: red).

**Figure 2 molecules-29-04237-f002:**
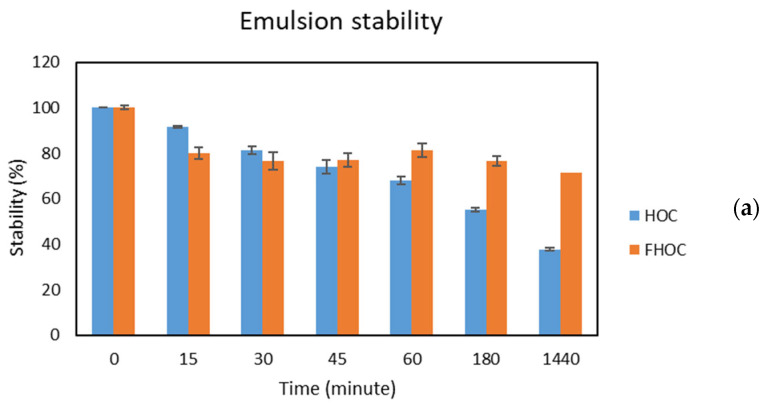
Emulsion stability (**a**) and foaming stability (**b**) of HOC and FHOC.

**Table 1 molecules-29-04237-t001:** Box–Behnken design of factors with experimental TPC values.

Run	Actual and Coded Levels of Factors			Response
	Initial Moisture Content (X_1_, %)	Incubation Temperature (X_2_, °C)	Incubation Time (X_3_, Day)	Total Peptide Content (mg Tryptone eqv./g DM of Solid Substrate)
1	70 (+1)	30 (0)	5 (+1)	369.8 ± 27.9
2	30 (−1)	20 (−1)	3.5 (0)	158.1 ± 24.5
3	50 (0)	30 (0)	3.5 (0)	346.3 ± 84.4
4	70 (+1)	40 (+1)	3.5 (0)	235.8 ± 81.8
5	30 (−1)	30 (0)	5 (+1)	113.9 ± 9.5
6	30 (−1)	40 (+1)	3.5 (0)	146.3 ± 77.0
7	50 (0)	20 (−1)	5 (+1)	344.0 ± 59.2
8	50 (0)	30 (0)	3.5 (0)	212.9 ± 38.5
9	50 (0)	30 (0)	3.5 (0)	280.9 ± 9.8
10	50 (0)	30 (0)	3.5 (0)	247.2 ± 66.1
11	30 (−1)	30 (0)	2 (−1)	132.1 ± 1.6
12	70 (+1)	30 (0)	2 (−1)	136.5 ± 84.5
13	50 (0)	40 (+1)	2 (−1)	104.6 ± 45.1
14	50 (0)	20 (−1)	2 (−1)	66.3 ± 32.3
15	50 (0)	40 (+1)	5 (+1)	191.8 ± 115
16	70 (+1)	20 (−1)	3.5 (0)	304.0 ± 139.1
17	50 (0)	30 (0)	3.5 (0)	307.9 ± 107.4

**Table 2 molecules-29-04237-t002:** Analysis of variance (ANOVA) for the fitted design model for optimization of SSF conditions.

Source	SS	DF	MS	F	*p*
Model	141,506	9	15,722.9	8.51	0.005
X_1_- Initial moisture content	35,841.8	1	35,841.8	19.41	0.003
X_2_- Incubation temperature	4703.0	1	4703.0	2.55	0.155
X_3_- Incubation time	48,013.0	1	48,013.0	26.00	0.001
X_1_X_2_	798.3	1	798.3	0.43	0.532
X_1_X_3_	11,200.7	1	11,200.7	6.06	0.043
X_2_X_3_	9070.3	1	9070.3	4.91	0.062
X_1_^2^	4664.4	1	4664.4	2.53	0.156
X_2_^2^	5067.7	1	5067.7	2.74	0.142
X_3_^2^	19,268.9	1	19,268.9	10.43	0.014
Residual	12,928.3	7	1846.9		
Lack of Fit	2181.9	3	727.3	0.27	0.844
Pure Error	10,746.4	4	2686.6		
Cor Total	154,434.3	16			
R^2^ = 0.916, Adjusted R^2^ = 0.809, Predicted R^2^ = 0.665					

DF: degrees of freedom, SS: sum of squares, MS: mean squares.

**Table 3 molecules-29-04237-t003:** Predicted and experimental values of TPC for model validation.

Run	A: Initial Moisture Content (%)	B: Incubation Temperature (°C)	C: Incubation Time (Day)	Predicted Total Peptide Content (mg Tryptone eqv./g DM of Solid Substrate)	Actual Total Peptide Content (mg Tryptone eqv./g DM of Solid Substrate)	Error (%)
1	67.3	25.8	4.3	379.6	358.9	5.8
2	64.4	20.4	4.6	391.3	354.1	10.5
3	69.8	24.6	4.6	401.2	404.4	0.8
4	70.0	30.0	5.0	375.4	366.1	2.5
5	60.8	24.3	4.5	370.0	425.9	13.1

**Table 4 molecules-29-04237-t004:** General characteristics of HOC and FHOC.

	Hazelnut Oil Cake	Fermented Hazelnut Oil Cake
Dry matter (%)	91.08 ± 0.02	85.43 ± 0.23
Water activity (25 °C)	0.733 ± 0.00	0.704± 0.00
Protein (%)	58.02 ± 0.77	64.64 ± 0.28
Lipid (%)	8.41 ± 0.12	9.19 ± 0.35
Ash (%)	6.05 ± 0.00	7.11 ± 0.0
Carbohydrate (%)	27.52	19.06
Color values		
L*	77.34 ± 0.03	54.83 ± 0.38
a*	2.39 ± 0.010	6.39 ± 0.07
b*	13.18 ± 0.05	16.47 ± 0.08

**Table 5 molecules-29-04237-t005:** Amino acid composition of HOC and FHOC (% amino acid/g cake).

Amino Acid (%)	HOC	FHOC
Aspartic acid (Asp) *	4.19 ± 0.18	5.45 ± 0.29
Glutamic acid(Glu) *	11.46 ± 0.35	12.53 ± 0.32
Serine (Ser)	1.44 ± 0.06	2.29 ± 0.13
Histidine (His)	1.01 ± 0.19	0.98 ± 0.18
Glycine (Gly)	2.06 ± 0.12	2.29 ± 0.28
Threonine (Thr)	1.22 ± 0.07	1.49 ± 0.05
Arginine (Arg) *	3.30 ± 0.40	8.21 ± 0.65
Alanine (Ala)	2.17 ± 0.14	2.62 ± 0.17
Tyrosine (Tyr)	1.53 ± 0.06	1.65 ± 0.17
Valine (Val)	3.15 ± 0.15	3.47 ± 0.12
Phenylalanine (Phe) *	1.76 ± 0.05	2.08 ± 0.04
Isoleucine (Ile)	1.49 ± 0.10	1.46 ± 0.08
Leucine (Leu)	2.94 ± 0.08	3.33 ± 0.22
Lysine (Lys)	1.80 ± 0.17	1.34 ± 0.28
Total *	39.55 ± 1.48	49.26± 1.10

Values are given as mean ± standard deviation. * There is a statistical difference between the results (*p* ≤ 0.05).

**Table 6 molecules-29-04237-t006:** Detected volatile compounds in HOC and FHOC (µg/kg dry weight).

LRI *	Compound	HOC	FHOC
921	3-Metyl butanal	-	298.7 ± 1.9
1080	Hexanal	605.8 ± 20.2	-
1130	2-Butylfuran	-	23.7 ± 0.3
1176	2-Heptanon	-	171.5 ± 4.5
1228	2-Pentylfuran	-	229.3 ± 9.6
1210	Limonene	1941.6 ± 193.8	-
1253	3-Octanone	-	160.0 ± 10.7
1245	1-Pentanol	293.1 ± 20.3	-
1273	Methylpyrazine	-	172.1 ± 13.2
1273	o-Simen	362.0 ± 10.7	-
1319	2,5-Dimethylpyrazine	-	565.3 ± 46.3
1364	1-Hexanol	309.0 ± 3.1	-
1395	2-Nonanone	-	1993.3 ± 131.6
1445	Acetic acid	2601.6 ± 154.7	-
1442	1-Octen-3-ol	-	377.8 ± 3.8
1445	1-Heptanol	106.0 ± 3.5	-
1487	2-Ethyl-1-hexanol	220.1 ± 11.6	50.6 ± 2.6
1506	Pyrole	-	571.4 ± 2.5
1525	Benzaldehyde	-	5537.6 ± 18.6
1571	2-Methylpropanoic acid	-	29.1 ± 0.4
1600	Benzonitrile	-	229.4 ± 2.1
1672	3-Methylbutanoic acid	-	520.4 ± 9.2
1731	Pentanoic acid	35.0 ± 2.3	-
1843	Hexanoic acid	142.1 ± 0.1	-
1868	Benzyl alcohol	-	84.4 ± 6.5
1890	Phenylethyl alcohol	-	42.1 ± 4.0
1934	Heptanoic acid	76.4 ± 0.1	-
2072	Octanoic acid	167.6 ± 13.7	-
2161	Nonanoic acid	132.7 ± 13.4	-
2079	p-Cresol	-	15.0 ± 1.0
2413	Benzoic acid	-	121.3 ± 6.5
2438	Indol	-	24.8 ± 0.7
2455	Benzophenon	-	141.4 ± 10.3

Values are given as mean ± standard deviation. * Linear retention index (LRI) values calculated for DB-Wax column.

**Table 7 molecules-29-04237-t007:** Bioactive properties of HOC and FHOC.

Bioactive Property	HOC	FHOC
Soluble protein content (mg BSA/g)	256.8 ± 3.3	279.5 ± 4.0
Total peptide content (mg tripton/g)	32.35 ± 7.76	462.37 ± 8.34
Total phenolic content (mg GA/g)	4.67 ± 0.02 *	16.17 ± 0.31 *
	0.12 ± 0.00 **	5.71 ± 0.29 **
	1.61 ± 0.02 ***	11.30 ± 0.39 ***
ABTS cation radical scavenging activity (µg trolox/g)	19.94 ± 0.41	75.61 ± 0.62
DPPH radical scavenging activity (µmol trolox/g)	2.90 ± 0.17	14.09 ± 0.24
OH radical scavenging activity (mg ascorbic acid/g)	239 ± 1	265 ± 1
α-Amylase inhibition activity (%)	N.D.	N.D.
α-Glucosidase inhibition activity (mg acarbose/g)	10.49 ± 0.12	182.11 ± 2.61
ACE inhibition activity (mg captopril/g)	1.79 ± 0.07	1.02 ± 0.16
AChE inhibition activity (%)	40.58 ± 0.77	40.73 ± 1.76

Values are given as mean ± standard deviation. * aqueous extract, ** methanolic extract, *** 80% methanolic extract (in water). N.D.: not detected. HOC: hazelnut oil cake, FHOC: fermented hazelnut oil cake, BSA: Bovine serum albumin, GA: Gallic acid, ABTS: 2,2′-azino-bis(3-ethylbenzothiazoline-6-sulfonic acid, DPPH: 2,2-diphenyl-1-picrylhydrazyl, OH: Hydroxyl, ACE: Angiotensin converting enzyme, AChE: Acetylcholinesterase.

**Table 8 molecules-29-04237-t008:** Functional properties of HOC and FHOC.

Functional Property	HOC	FHOC
Oil absorption capacity (g/g flour)	2.14 ± 0.06	3.31 ± 0.06
Water absorption capacity (g/g flour)	5.31 ± 0.06	2.53 ± 0.07
Emulsion activity index (m^2^/g)	42.32 ± 2.18	36.74 ± 1.42
Emulsion stability index (min) at 60 min	175 ± 10	763 ± 32
Foaming activity (mL)	10.96 ± 1.41	13.00 ± 0.88

**Table 9 molecules-29-04237-t009:** The investigated factors and their levels in the Box–Behnken experimental designs.

Factors	Unit	Sign	Actual Factor Levels		
			(−1)	(0)	(+1)
Initial moisture content	(%)	X_1_	30	50	70
Incubation temperature	(°C)	X_2_	20	30	40
Incubation time	(day)	X_3_	2	3.5	5

**Table 10 molecules-29-04237-t010:** The elution program in HPLC-DAD analysis.

Time (min)	Mobile Phase B (%)	Mobile Phase A (%)
0	2	98
3	2	98
26	57	43
28	100	0
35	100	0
35.5	0	98
40	0	98

## Data Availability

Data available on request due to restrictions.

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
