# Peer review of "Upgrading the Bioactive Potential of Hazelnut Oil Cake by Aspergillus oryzae under Solid-State Fermentation"

_molecules, 2024, doi:10.3390/molecules29174237_

Round 1

Reviewer 1 Report

Comments and Suggestions for Authors

The manuscript described a study about the Upgrading the Bioactive Potential of Hazelnut Oil Cake by Aspergillus Oryzae under Solid-State Fermentation. This work showed that the solid-state fermentation of hazelnut oil cake increased peptide content, soluble phenolic content, and antioxidant activity in the cake which had more oil absorption capacity with foaming and emulsion stability. The manuscript cannot be accepted in the valuable Journal of Molecules at the present form. The following comments (Major revisions) may be helpful to the authors for improving the manuscript:

-       The introduction needs to be rewritten to be more clear.

-       Add nomenclature to the manuscript.

-       The novelty of the work should be clearly stated.

-       Are all results, presented in Table 2 in the manuscript (Response Total peptide content (mg tryptone eqv./ g DM of solid substrate)), the average of different runs? If yes, mention the Standard deviation.

If no, use the updated literature review to confirm the obtained results.

-       Line 536 # …….due to the high extracellular enzymatic activities of Aspergillus oryzae#, are sure about the mentioned reasons? Please check, confirm, and give the reference related to your ideas and conclusion.

The same comment for Line 736 # …..due to the release of more functional groups with partial hydrolysis, and as a result of progressive fermentation, partially #.

 -       More recent references must be included to understand the Box-Behnken method. Given the journal's standards, the authors should ensure they cite high-quality, relevant journals. I suggest reading and add the following reference: (https://doi.org/10.1021/acsomega.0c03396).

-       I suggest the addition of some figures to the manuscript to become clear and more comprehensive (Determination of amino acid composition of HOC and FHOC can be presented in one figure…..).

-       The conclusion should be more quantitative.

-       Limitations, Recommendations and future work are missing.

The entire manuscript needs to be rewritten for better organization and improved English writing style.

Comments on the Quality of English Language

Minor editing of English language required

Author Response

Comment 1: The introduction needs to be rewritten to be more clear.

Response 1: We agree with the reviewer. We have revised, reorganized, and improved the introduction section of the manuscript. The relevant references were incorporated into the text. changes made in the text were highlighted in yellow.

Comment 2: Add nomenclature to the manuscript.

Response 2:  We have checked the "Instructions for Authors" section prepared by "Molecules" Journal of MDPI. Unfortunately, we could not incorporate the "nomenclature" into the text according to guidance. If the reviewer suggests and the Editors approve, we can prepare a nomenclature. 

Comment 3: The novelty of the work should be clearly stated

Response 3: We agree with the reviewer and appreciate for giving a chance us explain the novelty of the study better. We revised the text and clearly stated the novelty of the study in lines 120-122 and 147-151.

Comment 4: Are all results, presented in Table 2 in the manuscript (Response Total peptide content (mg tryptone eqv./ g DM of solid substrate)), the average of different runs? If yes, mention the Standard deviation. If no, use the updated literature review to confirm the obtained results.

Response 4: The standard deviation of the values was incorporated in Table 2. The results were obtained as the average of 2 repetitions of 2 replications.

Comment 5: Line 536 # …….due to the high extracellular enzymatic activities of Aspergillus oryzae#, are sure about the mentioned reasons? Please check, confirm, and give the reference related to your ideas and conclusion.  The same comment for Line 736 # …..due to the release of more functional groups with partial hydrolysis, and as a result of progressive fermentation, partially #.

Response 5: The related references were incorporated for the mentioned parts to Lines 601, and 751: Reference # 91;111, and 131. 

Comment 6: More recent references must be included to understand the Box-Behnken method. Given the journal's standards, the authors should ensure they cite high-quality, relevant journals. I suggest reading and add the following reference: (https://doi.org/10.1021/acsomega.0c03396).

Response 6: The mentioned reference was included in the manuscript. Reference # 64.

Comment 7:I suggest the addition of some figures to the manuscript to become clear and more comprehensive (Determination of amino acid composition of HOC and FHOC can be presented in one figure…..).

Response 8: Thank you very much for this suggestion. We prepared a figure consisting of amino acid compositions of HOC and FHOC but we realized that the increments and real values of the amino acids were not clearly visible from the graph. Therefore we have decided to proceed with the current table.

Comment 9: The conclusion should be more quantitative.

Response 9: The conclusion part was completely revised and rewritten with more quantitative results.

Comment 10: Limitations, Recommendations, and future work are missing.

Response 10: The suggested issues were mentioned in the conclusion part. We appreciate the reviewer for valuable suggestions that made our manuscript more quality.

Comment 11: The entire manuscript needs to be rewritten for better organization and improved English writing style.

Response 11: The whole manuscript was revised and many changes were made in the text. The changes were highlighted with yellow color.

Reviewer 2 Report

Comments and Suggestions for Authors

This study aimed to enhance the overall peptide yield in hazelnut oil cake through Aspergillus oryzae-mediated solid-state fermentation. To achieve this goal, the author optimized various process parameters and identified a significant model demonstrating that lower incubation temperatures coupled with longer durations and higher initial moisture levels led to increased total peptide production. It is an interesting work. Therefore, I would like to recommend its publication in this journal after the major revision with some issues to be carefully addressed.

1.     Error bars in Figure 2a and b need to be supplemented. In addition, the horizontal and vertical coordinates should present the name and unit separately.

2.     "µl" in line 316 and 317 needs to be modified to be consistent with the full text.

3.     Buffers of different pH are used during the determination of different substances in this manuscript. How is the pH of these buffers decided?

4.     Please confirm whether it is "100" or "100%" in Eq.2.

5.     The author did not clarify the source of reagents and instruments in the paper, which should be added.

6.     There are some grammatical errors in the paper, please check it carefully.

7.     The format of units in the whole paper has repeatedly been inconsistent, such as "80℃" in line 774 and "25℃" in line 765, which are different from other parts of the paper.

8.     It is advised to refine the abstract part to make the focus of the paper more intuitive for the reader.

Comments on the Quality of English Language

Moderate editing of English language required.

Author Response

Comment 1: Error bars in Figure 2a and b need to be supplemented. In addition, the horizontal and vertical coordinates should present the name and unit separately.

Response 1: We appreciate the reviewer point out the missing parts in the figures. Figures 2a and b were revised according to the reviewer's suggestion.

Comment 2: "µl" in line 316 and 317 needs to be modified to be consistent with the full text.

Response 2: The whole manuscript was revised and all the required corrections were made and highlighted with yellow color.

Comment 3:  Buffers of different pH are used during the determination of different substances in this manuscript. How is the pH of these buffers decided?

Response 3: The types of buffers were prepared according to the methods described in the cited references.

Comment 4: Please confirm whether it is "100" or "100%" in Eq.2.

Response 4: We checked the Eq 2 and confirmed the "100" as the current form in the manuscript.

Comment 5:  The author did not clarify the source of reagents and instruments in the paper, which should be added.

Response 5: We added the reagents in the materials part and included all the required information for the instruments in the methods section. The changes made were highlighted in yellow color. 

Comment 6: There are some grammatical errors in the paper, please check it carefully.

Response 6: We appreciate this valuable suggestion. We revised the whole manuscript and made some changes to improve the grammar of the language in the manuscript (highlighted in yellow).

Comment 7: The format of units in the whole paper has repeatedly been inconsistent, such as "80℃" in line 774 and "25℃" in line 765, which are different from other parts of the paper.

Response 7: The format of the symbols was checked and corrected throughout the manuscript. 

Comment 8: It is advised to refine the abstract part to make the focus of the paper more intuitive for

Response 8: The abstract was refined and shortened according to the instructions for authors as being a maximum of 200 words.

Round 2

Reviewer 1 Report

Comments and Suggestions for Authors

Thanks for you response, the manuscript can be accepted in the current form

Comments on the Quality of English Language

Minor editing of English language required.

Reviewer 2 Report

Comments and Suggestions for Authors

accept

  Comments on the Quality of English Language

accept